# Pesticide Exposure of Residents Living in Wine Regions: Protocol and First Results of the Pestiprev Study

**DOI:** 10.3390/ijerph20053882

**Published:** 2023-02-22

**Authors:** Raphaëlle Teysseire, Emmanuelle Barron, Isabelle Baldi, Carole Bedos, Alexis Chazeaubeny, Karyn Le Menach, Audrey Roudil, Hélène Budzinski, Fleur Delva

**Affiliations:** 1Bordeaux Population Health Research Center, Inserm UMR1219-EPICENE, University of Bordeaux, 33076 Bordeaux, France; 2Department of Occupational and Environmental Medicine, Bordeaux Hospital, 33000 Bordeaux, France; 3Regional Health Agency of Nouvelle-Aquitaine, 33076 Bordeaux, France; 4UMR 5805 EPOC, CNRS, Université de Bordeaux, 33400 Talence, France; 5ECOSYS, INRAE-AgroParisTech-Paris-Saclay University, 91120 Palaiseau, France

**Keywords:** fungicides, plant protection products, exposure assessment, indoor pollution, air pollution, agriculture

## Abstract

The PESTIPREV study has been designed to investigate residential exposure to pesticides applied to vines and ultimately propose mitigation measures. A feasibility study was carried out to validate a protocol for measuring six pesticides in three houses located near vineyards in July 2020. Samples included indoor and outdoor surfaces sampled with wipes (*n* = 214), patches on the resident’s skin (*n* = 7), hand or foot washing (*n* = 5), and pets sampled using wipes (*n* = 2). Limits of quantification for wipes ranged between 0.02 ng for trifloxystrobin and 1.50 ng for pyraclostrobin. Tebuconazole and trifloxystrobin were quantified in nearly 100% of the surface samples, whereas the other fungicides were less frequently found (from 39.7% for pyraclostrobin to 55.1% for boscalid). The median surface loadings ranged from 3.13 ng/m^2^ for benalaxyl to 82.48 ng/m^2^ for cymoxanil. The pesticides most frequently quantified in hand washing, patch samples, and pet wipes were the same as those quantified on surfaces. Finally, the analyses proved to be successful. The tools developed to collect information on determinants were well completed. The protocol was well received by the participants and appeared to be feasible and relevant to the objective of the PESTIPREV study, although some improvements have been identified. It was applied on a larger scale in 2021 to study the determinants of pesticide exposure.

## 1. Introduction

Epidemiological studies conducted among agricultural workers demonstrate that exposure to pesticides increases the risk of several adverse health conditions, such as cancer [1,2], neurological disorders [3,4], and adverse effects on reproductive organs [5]. People who live near fields sprayed with pesticides may be another group at risk of adverse health effects. According to previous studies, populations residing near agricultural areas would be more exposed to agricultural pesticides than the general population. The levels of compounds measured in domestic environments but also in biological samples tend to be higher in this population for several classes of pesticides (organophosphates, organochlorines, pyrethroids) [6,7].

However, the determinants of the residents’ exposure are poorly known [7]. Spatial indicators such as proximity to fields, crop acreage in the vicinity, and pesticides applied nearby have frequently been associated with elevated levels of pesticide exposure. These indicators confirm a significant contribution of the drift pathway in the overall exposure of individuals, resulting from aerial emissions of pesticides from the fields during application but also in the days after. Other determinants, including meteorological and topographical parameters, occupant characteristics and behaviors, hygiene practices, and the layout of the house, could explain exposure, but they have been less well documented or the findings of studies carried out in these areas have been conflicting. Knowledge of these determinants is crucial to improve the conduct of epidemiological and regulatory studies regarding the impact of pesticide exposure. This would also contribute to identifying measures to mitigate the exposure of populations living in agricultural areas.

France is one of the largest agricultural producers in Europe, with nearly 43.9% of the surface area being used for agriculture (278,142 km^2^) [8]. This means that many people live near agricultural crops treated with pesticides. In the Bordeaux region, vines represent the predominant crop and cover nearly 115,000 hectares, or half the utilized agricultural area (UAA) of the Gironde department [9]. Vineyards are closely intertwined with habitations [10] and the pesticide pressure associated with viticulture is particularly high [9].

In order to better characterize and understand the exposure of people living near vineyards, we conducted an observational study (the PESTIPREV study) in the Bordeaux region with the aim of identifying the determinants of exposure related to agricultural practices, meteorological and topographical conditions, the layout of buildings, and residents’ characteristics and behaviors. In order to assess the feasibility of this study, we first measured pesticides in three households located near vineyards. The objectives of this work were 1/to present the protocol implemented for this pilot study, 2/to describe the first results obtained, and 3/to identify the potential improvements for its deployment on a larger scale in a second phase.

## 2. Materials and Methods

### 2.1. Participants’ Enrollment

Three households were included in the feasibility study according to the following eligibility criteria: 1/the houses had to be located in the Bordeaux region, adjoining one or several vineyards treated with pesticides (no minimum or maximum distance required); 2/the people in the households were not employed in jobs involving manufacturing or handling pesticides, as previous publications have demonstrated the significant contribution of the take-home exposure pathway in families of farmers [11,12]; and 3/participants had to provide informed consent to take part in this study.

Potential participants selected from the research team’s contacts were sent an e-mail outlining the objectives and procedures of the study. An investigator then called the volunteers to provide detailed information about taking part and to schedule a half-day home visit during the most frequent pesticide application period.

### 2.2. Data Collection

#### 2.2.1. Questionnaires

Prior to the visit, subjects were given a calendar to fill in with their locations and activities for the two days preceding the home visit. On the day of the visit, the calendar was collected, and a standardized questionnaire was administered to one adult in the house by an investigator after having collected the consent forms from all participants. The questionnaire, designed for the PESTIPREV study, recorded the characteristics of the household members (age, gender, employment, and education), the layout of the house (number of floors, rooms, position of the doors and windows, and the state of renovation of the house), hygiene habits (housekeeping, the presence of pets, and domestic use of pesticides), outdoor activities (gardening, exercise or play, eating, and relaxing), and the dates of the last known agricultural spraying in the nearby vineyards.

#### 2.2.2. Surface Wipe Samples

Indoor and outdoor pesticide residues were sampled on a range of surfaces using wipes. We collected samples from six areas of the home: the garden, the entrance to the house, the living room, the kitchen, the parents’ bedroom, and the child/children’s bedroom. Investigators had the option of adding an additional area to the sampling plan if necessary. Inside the home, we sampled surfaces considered to reflect indoor pesticide contamination over several weeks/months (high horizontal surfaces such as shelves, doors, or window frames). We hypothesized that, as had been demonstrated for carpet dust [13,14], these surfaces were protected from sunlight and microbial degradation, and were likely to accumulate particles over a prolonged period. We also selected surfaces that might reflect shorter-term exposure (such as floors and furniture) and a series of frequently touched surfaces that might represent the handling of pesticides (such as light switches, decorative objects, toys, and digital equipment). Outside, we sampled surfaces characterizing the most recent local deposits of pesticides (floors and windows), as well as furniture or gardening tools, representing potential exposure to pesticides during outdoor activities. We also sampled pet fur using wipes.

Non-woven sterile cotton gauze wipes (10 × 10 cm, 4-ply, 30 g/m^2^, LCH^®^, Hong Kong, China) moistened with 5 mL ethanol (analytical grade) were used for wipe surface samples. The operator wore nitrile gloves, which were changed between each sampling. For the majority of surfaces, a single wipe was used for sampling. Flat surfaces (maximum 1 m^2^) were wiped in an S-shape. The operator moved the wipe in overlapping strips from side to side until the surface to be sampled was covered. A second pass was made after folding the wipe in two (contaminated part inside) in the same way but perpendicularly. A second folding was conducted before placing the wipe in a glass bottle. Sampled surfaces were measured by the investigators using a template, a caliper, or a measuring tape. Field blanks were prepared by exposing moistened gauze in situ for 1 min to air.

During sampling, an investigator compiled observational data on the selected surfaces (in terms of their size, material, how often they were used, and when and how they were last cleaned) in a structured observation notebook, with input from household members if necessary.

All wipes were individually placed in separated, cleaned glass bottles and placed in an electric cooler at −18 °C immediately after sampling. They were transported to the laboratory within four hours and stored at −20 °C before analysis [15].

#### 2.2.3. Individual Measures: Hand Washing and Patches

Individual measurements of external contamination were performed on at least one person per household, including hand washing and forearm measurements (patches).

Hand washing was performed during the home visits and the times of the current and previous hand washes were recorded. Participants were asked to place their hands over a clean stainless-steel basin. They then rubbed their hands while an investigator slowly poured 500 mL of mineral water over them. The contents of the basin were then poured into a glass bottle.

At the beginning of the visit, patches made of cotton gauze lined with aluminum foil and medical tape were placed on the participant’s right forearm, as described by Bureau et al., 2022 [16]. We also placed additional patches on both legs of one subject wearing a short skirt. The patches were removed at the end of the visit and the times were recorded. The wipes were then placed in glass bottles.

Subjects told the investigator what activities they had done between their last hand washing and this one, as well as while the patch was applied.

All of the samples (wipes and water) were placed in an electric cooler at −20 °C for immediate storage and transport after sampling. Samples were then stored in the laboratory at −18 °C prior to analysis.

### 2.3. Pesticide Analysis

#### 2.3.1. Pesticides Selected

Active substances were selected a priori according to several criteria. First, they had to be conventional pesticides (without microorganisms or natural products) authorized for use in France in 2019 in the wine industry as mentioned in the European Union pesticide database [17] and in the French E-Phy database [18]. In total, 75 active substances met these criteria.

The active substances were then prioritized based on the tonnages sold in the Gironde department in 2017, the date of the last update of the National Bank of Agricultural Crop Protection Product Sales (BNVD—Banque nationale des ventes de produits phytosanitaires agricoles) [19]. The frequency of application during the year and their duration of use, as determined by expert opinion, were taken into account. In the final selection phase, we used analytical criteria to select the six substances: benalaxyl, boscalid, cymoxanil, pyraclostrobin, tebuconazole, and trifloxystrobin. These pesticides are not intended for domestic use, except for tebuconazole, which is also used to protect wood against wood-eating insects [20]. The main information about properties and use of the six pesticides, obtained from the Pesticide Properties Database (PPDB) website [21] and the BNVD [19], is presented in Table 1.

#### 2.3.2. Quantitative Analysis of Pesticide Residues

Wipe samples were ultrasonically extracted using acetonitrile (two sequential extractions with 50 mL). Internal standards were added prior the extraction. An amount of 10 mL of solvent was recovered and reconcentrated to 1 mL. The final extracts were analyzed by LC/MS/MS using Multiple Reaction Monitoring mode (MRM).

Artificial spiked samples (with clean wipes) were added for each series of extraction and were processed in the same way as wipe samples. Overall, recoveries for the validation samples were comprised between 80 and 110% with variabilities below 20%. Signal-to-noise ratios (S/N) were determined using the peak-to-peak method and LOQs were calculated for S/N = 10 (LOQspi).

Laboratory blanks for both types and samples were performed and analyzed using the same procedure. A few pgs of benalaxyl, tebuconazole, and pyraclostrobin were found in some series. Levels in the samples were blank-subtracted. Limits of quantification (LOQblk) were calculated taking into consideration the presence in the blanks when relevant (i.e., benalaxyl, tebuconazole, pyraclostrobin) targeting the maximum quantities. In this case, they were calculated as five times the level found in the blanks (LOQblk).

Limits of quantification (LOQ) in wipes (surfaces and patches) were chosen as the maximum between blank LOQ (LOQblk) and LOQ calculated with spike samples (LOQspi). They were inferior to 1.50 ng per sample: LOQ benalaxyl = 0.06 ng, LOQ boscalid = 0.06 ng, LOQ cymoxanil = 1.00 ng, LOQ pyraclostrobin = [0.70 ng–1.50 ng], LOQ tebuconazole = 0.05 ng, LOQ trifloxystrobin = 0.02 ng. The surface areas ranged in size from 5.47 × 10^−4^ to 1 m^2^, meaning that the limits of quantification per surface area were highly variable, ranging from 1.1 × 10^−5^ ng/m^2^ for trifloxystrobin to 1 ng/m^2^ for cymoxanil. Only three interferences were observed (for boscalid in two samples and for tebuconazole in one).

For hand washing, water samples (after addition of internal standards) were directly injected after filtration at 0.2 mm and analyzed by LC/MS/MS (MRM mode) using internal standard methodology for quantification. Spike waters (with ultra-pure water) and laboratory blanks were performed. No compound was found in the blanks and the overall recoveries were satisfactory, ranging between 92% and 105% with variability lower than 15%. Limits of quantification (LOQ according to S/N > 10) in hand washing samples were LOQ benalaxyl = 0.2 ng/L, LOQ boscalid = 8.0 ng/L, LOQ cymoxanil = 60.0 ng/L, LOQ pyraclostrobin = 0.4 ng/L, LOQ tebuconazole = 0.7 ng/L, LOQ trifloxystrobin = 0.5 ng/L. Details of the analysis method developed for the PESTIPREV study are available in the Appendix A.

### 2.4. Data Management and Statistical Analysis

The information collected during the home visit and the results of the pesticide analysis were entered into a database; descriptive statistics were performed using R version 4.1.2 [22].

For all surface samples and for each substance, we calculated the detection and quantification frequencies per household and for all houses combined. We also calculated the surface loading defined as the quantity of pesticides related to the surface area, expressed as mass per unit area (ng/m^2^). For individual measurements, we considered the mass of pesticides quantified per volume (liter of water) or per sampling time (patches).

Next, we plotted the distribution of the surface loadings per pesticide based on several parameters: location in the house, type of surface sampled, and the time and method of the last cleaning. Pesticide amounts below the detection limit (*n* = 3) or those not calculated due to analytical interferences (*n* = 3) were considered missing data. For pesticides that were detected but not quantified, we imputed a surface loading value equal to half the limit of quantification divided by the surface area. As the matrices presented different surface areas, this substitution method was not expected to strongly affect the distribution of the data.

### 2.5. Assessment of Acceptability

In order to assess the acceptability of our protocol, an interview was scheduled with a member of the household who was present during the initial visit a few months after the data collection. The semi-structured interview was conducted by telephone using a guided questionnaire consisting of nine items. It included brief questions about the planning and execution of the home visits (organization, duration, and participant involvement), potential disruptions during the visits or sampling, whether the study was perceived to be useful, reasons for the study, and reproducibility of the protocol. An open-ended question about possible provisions that could improve the participation of future subjects was also included at the end of the interview.

## 3. Results

### 3.1. Population and Household Characteristics

Three households were included in the PESTIPREV 1 study, with visits conducted between 8 and 16 July 2020. The main characteristics of the households are reported in Table 2.

Houses 1 and 3 were located on flat land and had discontinuous hedgerows separating the garden from the nearest vineyards. Dwelling 2 was located on a hillside and had no hedgerow.

The habitants of the first household were two retired adults and a student, while the other two households were home to working people and their children. All but one of the adults had more than two years of post-high school education.

### 3.2. Detecting and Quantifying Pesticides in Samples

#### 3.2.1. Pesticides on Indoor and Outdoor Surfaces

For all of the homes, we sampled the garden, the main entrance, the living room, the kitchen, and one adult bedroom. Two children’s bedrooms were also sampled, in houses 2 and 3. A second adult bedroom and a playroom were sampled in houses 1 and 2, respectively.

A total of 214 surface samples were collected: 67, 69, and 78 samples for houses 1, 2, and 3, respectively.

The sampled surfaces included frequently touched surfaces (*n* = 75, 25.2%), various exterior surfaces (*n* = 30, 14.0%), overhead surfaces (*n* = 26, 12.1%), furniture (*n* = 18, 8.4%), exterior windows (*n* = 14, 6.5%), floors (*n* = 13, 6.1%), interior windows (*n* = 12, 5.6%), and other surfaces (*n* = 27, 12.6%).

We detected all of the pesticides on all surfaces, except for cymoxanil in three samples from surfaces inside and outside house 3. The frequency of quantification by pesticide and the surface loadings for the three houses are presented in Table 3 (for the quantified pesticides only). Tebuconazole and trifloxystrobin were quantified in almost 100% of the surface samples. The other four fungicides were less frequently quantified, with some mixed results between houses: while cymoxanil was quantified in only 6.4% of the samples in house 3, it was measured on 95.5% of the surfaces in house 1. The median of the surface loadings was between 2.73 ng/m^2^ for benalaxyl in house 3 and 148.26 ng/m^2^ for boscalid in house 1.

A figure representing the frequency of quantification per pesticide and the median surface loading for each home according to the type of surface sampled is provided in the Appendix A. We observe that quantification rates and concentrations tend to be higher for raised surfaces and frequently touched surfaces and lower for windowpanes and floors.

#### 3.2.2. Individual Measurements

We collected four hand-wash waters (from four individuals), one foot-wash water (from one individual), and seven patches (from five individuals). All of the pesticides were detected in all samples. The results of these analyses and the limits of quantification for each pesticide are presented in Table 4 and are represented in the Appendix A. Tebuconazole and trifloxystrobin were quantified in nearly 100% of individual measurements. Cymoxanil and benalaxyl were found once in hand-wash water and cymoxanil was quantified twice in the patches. The highest levels of cymoxanil in the waters and patches and the highest levels of tebuconazole in the patches were from the adult in house 1. The highest concentrations of benalaxyl (hand-wash water) and the highest levels of tebuconazole and trifloxystrobin in the patches were from a child in house 3.

The adult in house 1 reported gardening and hanging laundry out to dry. The other three adults reported no specific activities. The two children reported a variety of activities such as eating and playing inside or outside the house, and the children in house 3 reported playing with the dog.

#### 3.2.3. External Contamination of Pets

During the home visits, one cat from house 1 and one dog from house 3 were sampled with wipes. In both cases, participants reported that their pets usually spent more than an hour outside per day, including in the vineyards next to the house. All of the substances were detected in both wipe samples. The pesticides quantified on the cat were cymoxanil (3.10 ng), tebuconazole (0.13 ng), and trifloxystrobin (0.06 ng). The pesticides quantified on the dog were boscalid (0.85 ng), tebuconazole (1.79 ng), and trifloxystrobin (0.06 ng).

### 3.3. Determinants of the Pesticide Surface Loadings on the Surfaces

The distribution of pesticide surface loadings by location in the house and garden is shown in Figure 1. The highest medians of surface loadings were found in the front entrance for all pesticides except tebuconazole, for which the highest median was found in a child’s bedroom. The lowest medians of surface loadings were found outside for pyraclostrobin and tebuconazole, in the living room for boscalid and cymoxanil, and in a playroom for benalaxyl and trifloxystrobin.

Figure 2 shows the distribution of pesticide surface loadings according to the type of surfaces sampled inside the home. For all pesticides, the lowest median surface loadings were found on interior windowpanes and the highest surface loadings were found for overhead surfaces, except for cymoxanil. The highest median cymoxanil concentrations were found on frequently touched objects (handled objects, switches, door handles, etc.)

Figure 3 shows the distribution of concentrations based on the last time the surface was cleaned. The highest medians of surface loadings for benalaxyl, pyraclostrobin, tebuconazole, and trifloxystrobin were found for surfaces last cleaned more than six months ago, and for surfaces cleaned within the previous month for boscalid and cymoxanil. For all of the pesticides, the lowest median concentrations were found for surfaces cleaned less than a week ago.

For all pesticides, concentrations were higher in the areas reported as never or rarely cleaned, with concentrations ranging from 2.52 ng/m^2^ for pyraclostrobin to 64.4 ng/m^2^ for tebuconazole. For boscalid, pyraclostrobin, tebuconazole, and trifloxystrobin, the highest median surface loadings were found on surfaces that were wiped clean (8.50, 0.75, 18.4, and 9.48 ng/m^2^ respectively). Lower levels of benalaxyl and cymoxanil were found on surfaces cleaned with a vacuum cleaner (median 0.456 ng/m^2^ and 5.33 ng/m^2^).

### 3.4. Acceptability

Three interviews were conducted between February and March 2021. All respondents found the study useful. Participants were satisfied with how the study was organized and were not inconvenienced by the presence of the investigators in their homes. One person found the process to be relatively long, but appropriate for the number of samples collected. All participants felt that the format of this protocol was reproducible but stressed the importance of pre-visit explanations to ensure that it would be well understood and received by future participants.

The notebook and questionnaire given to participants were well completed, and they allowed us to collect information on potential exposure determinants. Unfortunately, the calendar was not adequately completed by participants and could not be used to explore the determinants of pesticide surface loadings on surfaces.

## 4. Discussion

### 4.1. Protocol Validity

The main objective of this pilot study was to evaluate the validity of a protocol for measuring the external exposure to pesticides of people living near vineyards and for collecting information on a series of potential exposure determinants.

The protocol implemented in 2020 at the first three sites proved to be mostly suitable for this purpose. The measurements allowed us to collect a large number of samples from surfaces (*n* = 214), patches (*n* = 7), hand or foot washing water (*n* = 5), and domestic animals (*n* = 2). The laboratory analysis proved successful in measuring the six targeted pesticides. The tools used to collect information on the determinants during the survey (questionnaire, notebook, etc.) were satisfactory in terms of completion rate. Finally, the protocol was well received by the participants, mainly due to the information provided before the survey on how it would be organized and what it would involve for the participants.

Nevertheless, the initial results indicate that adjustments to the protocol would be beneficial, mainly for logistical reasons, as well as to ensure that participants are well prepared for and fully understand the study.

First, the method of selecting participants should be adapted. Identifying eligible housing in the area using a Geographic Information System (GIS) and then making contact by phone or letter could enable us to include a wide range of profiles. As the 2020 participants pointed out, it is important that participants are well informed from the outset in order to ensure that they understand and follow our protocol.

Next, the sampling strategy will need to be reviewed. The number of samples per wipe per house should be limited and the type of surfaces sampled should be standardized to ensure better comparability between houses and thus sufficient statistical power to study the potential exposure determinants. Improvements could be made by avoiding sampling porous or rough surfaces and focusing on flat, geometrically well-defined surfaces. Sampling of very small areas should also be avoided, as this greatly increases the sensitivity of the measurement. Patches were only worn during the short period of the study and people did not perform their usual activities while the investigators were in their homes. Therefore, the use of patches could be discontinued.

The packaging of the wipes could be adapted to reduce their volume and minimize the use of materials to facilitate their transportation and storage until they are analyzed in the laboratory. The replacement of glass bottles by aluminum foil could also be considered. Regarding laboratory analyses, developments will be made in order to measure additional molecules such as folpel, which represents one of the most applied compounds in vineyards.

Finally, while the questionnaire and the notebook were suitable for collecting information, the calendar was difficult for participants to fill out and did not allow us to collect precise information on the activities performed before the home visit. Therefore, the calendar could be replaced by new items in the questionnaire.

Our protocol allows us to collect a very large number of samples per house, in several areas of the home. This makes it a high spatial resolution study. This format seems particularly well adapted to the detailed study of the determinants of residential external exposure (in terms of the sample, the room, and the home). Our study is complementary to other studies on this subject. These studies typically include fewer environmental measurements per household and often focus on indoor air or house dust collected with a vacuum cleaner [23,24,25]. These measurements are complemented by impregnation measurements (most often in urine). These study plans are well suited to compare pesticide exposure levels between different groups (residents/non-residents), to study individual determinants of internal exposure, or to explore certain general determinants of external exposure at home.

### 4.2. Presence and Surface Loading of Pesticides in the Houses

All six pesticides were detected in nearly all surface samples collected from the homes, with a detection rate close to 100%. The frequencies of quantification and medians of surface loadings per pesticide for all homes were as follows: benalaxyl (54.7%, 3.13 ng/m^2^), boscalid (55.1%, 47.55 ng/m^2^), cymoxanil (40.7%, 82.48 ng/m^2^), pyraclostrobin (39.7%, 3.40 ng/m^2^), trifloxystrobin (100%, 20.22 ng/m^2^), and tebuconazole (99.5% 38.05 ng/m^2^).

Previous observational studies rarely sampled the pesticides we included on residential surfaces using wipe sampling. We identified one relevant French study conducted by Beranger et al. in 2012, which was conducted in 239 homes in the Rhône-Alpes region, including 68 sites located near vineyards [26]. The authors analyzed the prevalence of 276 pesticides in recent dust obtained with a dust trap or wipes on the ground and in settled dust collected with wipes on the top edges of doors or window frames. For the houses located near vineyards, cymoxanil, tebuconazole, and trifloxystrobin were detected in both recent dust (in 49, 56, and 16% of the houses, respectively) and in settled dust (in 9, 54, and 12% of the houses, respectively). Benalaxyl was detected only once in settled dust. Boscalid and pyraclostrobin were excluded from the results. Detection limits were not available, which makes it difficult to compare with our data.

For the first three homes we studied, the difference in quantification frequencies observed between pesticides appears to be largely explained by the variable performance of the analysis for each compound. Tebuconazole and trifloxystrobin, which had the lowest LOQs, were quantified in 100% of the surface samples, while pyraclostrobin and cymoxanil had higher LOQs and were measured in 40% of the wipe samples.

On the contrary, the surface loadings seem to be better explained by the quantities of pesticides sold in Gironde in 2019 (shown in Table 1), since we found the highest concentrations for the most commonly sold pesticides (cymoxanil, tebuconazole, trifloxystrobin). However, this was not the case for boscalid, which showed relatively high concentrations compared to the average tonnage sold. The high dose of this pesticide applied per hectare or its long half-life could be explanatory factors in this case.

The results of the surface samples per home provided additional information. The first house had a different contamination profile with a higher detection frequency and surface loading of cymoxanil (and boscalid to a lesser extent) than the other two houses. This suggests a possible influence of local spraying that cannot be confirmed by knowledge of nearby farming practice.

The results for pesticides quantified in water, patches, and pets were consistent with those obtained on household surfaces. The pesticides quantified in these samples corresponded to the pesticides most frequently quantified on inert housing surfaces. In addition, high concentrations of cymoxanil were found in the samples collected from the first house. The gardening activities performed by the participant prior to hand washing could explain the high concentrations found on her hands. These trends and the underlying hypotheses that we have formulated based on these initial results will need to be confirmed in the second phase of the study.

### 4.3. Determinants of the Pesticide Levels on the Surfaces

For all of the pesticides, the median surface loading was, in most cases, lower outdoors than in the various rooms of the house. These results suggest that, while pesticide contributions from local spraying play an important role, the elimination of molecules due to rainfall or biological and physical degradation processes must also be taken into account. Inside the home, these substances can easily accumulate on surfaces that are protected from sunlight, moisture, temperature changes, or other physical factors.

The highest medians of pesticide surface loadings were recorded in the main entrance, which represents a major traffic area between the inside and the outside of the home. These results are consistent with those of Nishioka et al., who found that pesticide tracking from an active dog and from the homeowner were the most significant factors in 2,4-dichlorophenoxyacetic acid (2,4-D) intrusion inside residences after lawn treatments [27].

Elevated surfaces (shelves, doors, and window frames) and hand-held objects (digital equipment, toys, switches, etc.) appeared to be more contaminated with pesticides than other surfaces. This may reflect the influence of cleaning, as these types of surfaces are generally not cleaned often. Inside the house, we also observed that the most recently cleaned surfaces had the lowest surface loading measurements. This is consistent with the results of previous studies of populations living in agricultural areas [28,29,30] or in farmers’ houses [31,32], which indicated that frequent cleaning could limit the contamination of the home or the exposure of its occupants. The amount of data was not sufficient to observe clear trends on the most effective cleaning method.

All of these observations were conducted by varying one parameter at a time without considering other potential determinants. Therefore, no solid conclusions can be drawn from these first results. They have, however, allowed us to construct hypotheses that will have to be confirmed in the second phase of the PESTIPREV study. We will explore the impact of spatial determinants (proximity to fields and intensity of agricultural activities in the area) and non-spatial factors (household characteristics, layout of the building, hygiene habits, and the occupants’ activities) on exposure to pesticides. Knowledge of the determinants that contribute significantly to pesticide exposure is necessary in order to implement preventive measures for populations living in agricultural areas. Thus, identifying the determinants would enable us to identify households at risk of exposure and to propose preventive measures at home or even mitigation measures at a higher level.

## 5. Conclusions

The PESTIPREV protocol implemented in three households in 2020 was deemed appropriate for measuring pesticide exposure of people living near vineyards and can be replicated on a larger scale subject to a few adjustments. This feasibility study allowed us to describe the detection and quantification frequencies of six pesticides frequently applied to vineyards as well as their concentrations in different environments. In addition to analytical performance, certain parameters seemed to influence the concentrations of pesticides found on the surfaces sampled in the home, such as the quantities of pesticides sold in the area, the type of surface sampled, the room itself, and the cleaning habits. These trends will need to be confirmed in the second phase of the PESTIPREV study, which will be conducted in approximately 30 homes. The new data obtained will allow us to study the impact of spatial determinants (proximity to fields and intensity of agricultural activities in the area) and non-spatial factors (household characteristics, building layout, hygiene habits, and the occupants’ activities) on pesticide exposure.

## Figures and Tables

**Figure 1 ijerph-20-03882-f001:**
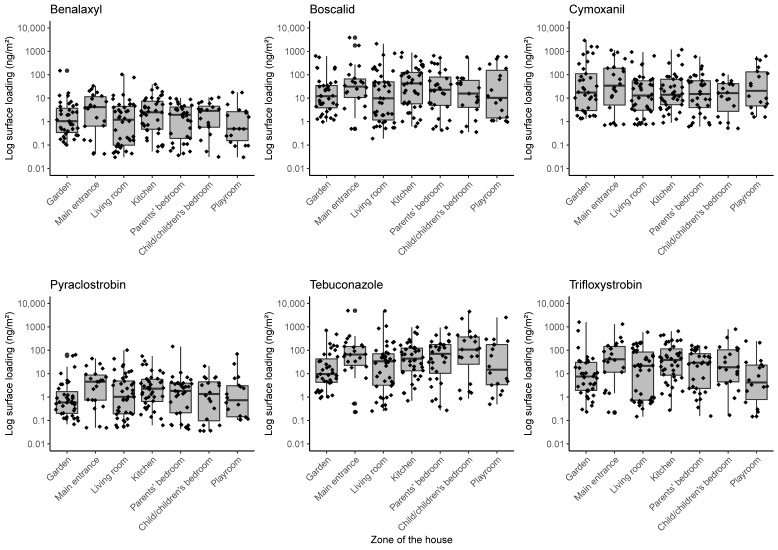
Log—pesticide surface loadings (ng/m^2^) according to the zone sampled (N = 214).

**Figure 2 ijerph-20-03882-f002:**
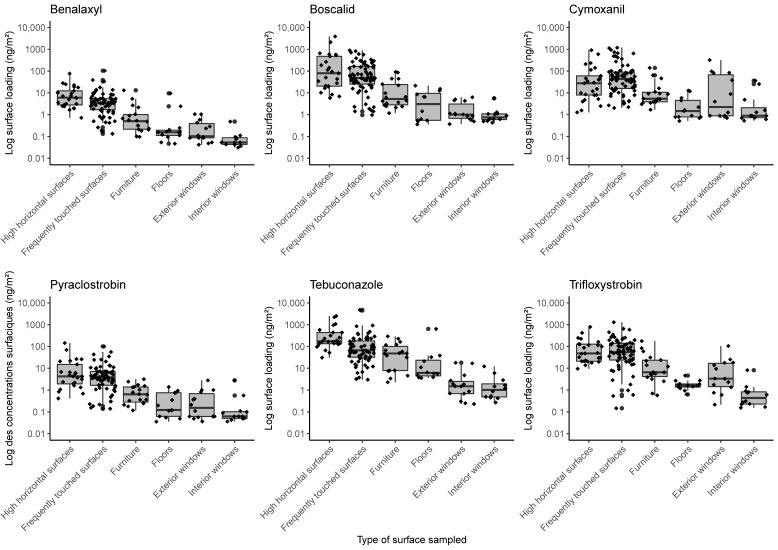
Log—pesticide surface loadings (ng/m^2^) according to the type of surface sampled (N = 214).

**Figure 3 ijerph-20-03882-f003:**
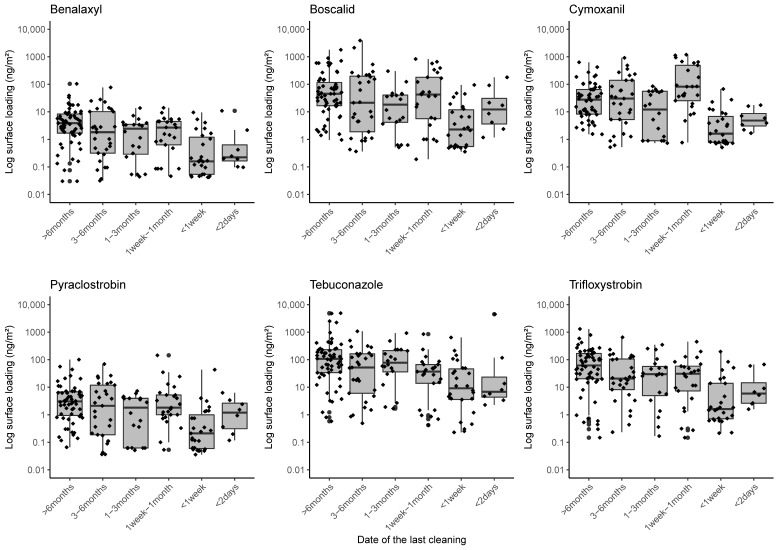
Log—pesticide surface loadings (ng/m^2^) according to the time of the last cleaning (N = 214).

**Table 1 ijerph-20-03882-t001:** Properties and uses of pesticides analyzed in the PESTIPREV pilot study.

Substance	Benalaxyl	Benalaxyl-M	Boscalid	Cymoxanil	Pyraclostrobin	Tebuconazole	Trifloxystrobin
CAS Number	71626-11-4	98243-83-5	188425-85-6	57966-95-7	175013-18-0	107534-96-3	141517-21-7
Molecular mass	325.40	325.40	343.21	198.18	387.82	307.82	408.37
Pv (mPa) ^a^	5.72 × 10^−1^	5.95 × 10^−2^	7.2 × 10^−4^	1.5 × 10^−2^	2.60 × 10^−5^	1.30 × 10^−3^	3.40 × 10^−3^
DT_50_ soil (days) ^b^	33.2	98.5	484.4	1.7	41.9	365.00	0.34
Use ^c^	FU	FU	FU	FU	FU	FU, PG	FU
Examples of targets (vines)	blue mold, late blight, downey mildew	downy and powdery mildew, late blight, grey mold	powdery mildew, grey mold, botrytis	downy and powdery mildew	black-rot, mildew	powdery mildew, black spot	black root, black spot, botrytis, downy and powdery mildew, leaf spot
Indicative dose applied on vine (g/ha)	150	75	600	90	100	108	65
Quantities sold in Gironde in 2019 (kg)	948	84	1093	4314	1797	3452	4407

^a^ Vapor pressure at 20 °C (mPa), ^b^ Aerobic Soil Half-Life (avg, days) (lab at 20 °C), ^c^ FU: fungicide, PG: Plant Growth regulator.

**Table 2 ijerph-20-03882-t002:** Characteristics of the households included in the PESTIPREV 1 study.

		Dwelling 1	Dwelling 2	Dwelling 3
Subjects				
	Adults	3	2	2
	Children	0	2	1
Construction date		1970s	<1900s	<1900s
Total surface (m^2^)		126	100	125
Home layout				
	Number of rooms	7	6	7
	Number of floors	2	1	1
Wood treatment in the last 5 years		0	0	0
Home renovations in the last 6 months		yes	no	yes
Number of pets (dogs, cats)		1	0	1
Distance between home and nearest field (m)		18	7	5
Time since the last spraying (days)		0	6	NA

**Table 3 ijerph-20-03882-t003:** Pesticides on surfaces in the three dwellings: quantification frequency and distribution of the surface loadings.

	Quantification of Pesticides		Surface Loading of Quantified Pesticides (ng/m^2^)	
	Occurrence (*n*)	Quantification Frequency (%)	Median	(Q25–Q75)
All surface wipes (*n* = 214)				
Benalaxyl	117	54.7	3.13	0.90–9.23
Boscalid	118	55.1	47.55	15.47–193.53
Cymoxanil	87	40.7	82.48	21.54–283.20
Pyraclostrobin	85	39.7	3.40	1.25–7.30
Tebuconazole	213	99.5	38.05	5.57–133.70
Trifloxystrobin	214	100	20.22	2.32–68.83
Dwelling 1 (*n* = 67)				
Benalaxyl	34	50.7	3.83	0.48–11.80
Boscalid	50	74.6	148.26	22.48–283.25
Cymoxanil	64	95.5	100.45	32.28–435.66
Pyraclostrobin	26	38.8	2.81	0.64–9.49
Tebuconazole	67	100	30.39	3.07–92.84
Trifloxystrobin	67	100	17.40	2.02–51.20
Dwelling 2 (*n* = 69)				
Benalaxyl	43	62.3	3.15	1.12–8.87
Boscalid	35	50.7	44.83	18.74–140.47
Cymoxanil	18	26.1	51.45	9.46–100.42
Pyraclostrobin	28	40.6	3.28	1.72–13.16
Tebuconazole	69	100	44.25	9.57–133.70
Trifloxystrobin	69	100	31.31	8.15–137.36
Dwelling 3 (*n* = 78)				
Benalaxyl	40	51.3	2.73	0.86–7.43
Boscalid	33	42.3	30.30	6.17–74.14
Cymoxanil	5	6.4	37.18	19.76–40.37
Pyraclostrobin	31	39.7	3.68	1.19–6.18
Tebuconazole	77	98.7	49.17	5.34–163.26
Trifloxystrobin	78	100	11.82	1.49–44.74

Benalaxyl (LOQ = 0.06 ng), Boscalid (LOQ = 0.06 ng), Cymoxanil (LOQ = 1.00 ng), Pyraclostrobin (LOQ = [0.70 ng–1.50 ng]), Tebuconazole (LOQ = 0.05 ng), Trifloxystrobin (LOQ = 0.02 ng).

**Table 4 ijerph-20-03882-t004:** External contamination of participants: pesticide amounts in hand washing and foot washing waters (*n* = 5) and in patches (*n* = 7).

	Hand Washing and Foot Washing Waters (*n* = 5)					Patches (*n* = 7)						
Dwelling	1	2	2	3	3	1	2	2	2	3	3	3
Subject	adult	child	child	adult	child	adult	adult	adult	adult	adult	adult	child
Localization ^a^	both hands	both hands	both feet	both hands	both hands	forearm R	forearm R	leg R	leg L	forearm R	forearm R	forearm R
Time since the last washing (min)	195	420	300	30	960							
Quantities (ng/h) ^b^												
Benalaxyl	<LOQ	<LOQ	<LOQ	<LOQ	0.17	<LOQ	<LOQ	<LOQ	<LOQ	<LOQ	<LOQ	<LOQ
Boscalid	<LOQ	<LOQ	<LOQ	<LOQ	<LOQ	<LOQ	<LOQ	<LOQ	<LOQ	<LOQ	<LOQ	<LOQ
Cymoxanil	399.69	<LOQ	<LOQ	<LOQ	<LOQ	2.96	<LOQ	0.28	<LOQ	<LOQ	<LOQ	<LOQ
Pyraclostrobin	<LOQ	<LOQ	<LOQ	<LOQ	<LOQ	<LOQ	<LOQ	<LOQ	<LOQ	<LOQ	<LOQ	<LOQ
Tebuconazole	8.33	1.69	3.94	3.48	5.28	0.07	0.09	0.03	0.03	0.13	0.21	0.58
Trifloxystrobin	0.81	<LOQ	3.24	1.89	2.83	0.02	0.02	0.03	0.02	0.03	0.09	0.18

^a^ R: right, L: left. ^b^ Limits of quantification in hand washing and foot washing waters: benalaxyl (LOQ = 0.2 ng/L), boscalid (LOQ = 8.0 ng/L), cymoxanil (LOQ = 60.0 ng/L), pyraclostrobin (LOQ = 0.4 ng/L), tebuconazole (LOQ = 0.7 ng/L), trifloxystrobin (LOQ = 0.5 ng/L). Limits of quantification in patches: benalaxyl (LOQ = 0.06 ng), boscalid (LOQ = 0.7 ng), cymoxanil (LOQ = 1.0 ng), pyraclostrobin (LOQ = 0.07 ng), tebuconazole (LOQ = 0.05 ng), trifloxystrobin (LOQ = 0.02 ng).

## Data Availability

The datasets generated and/or analyzed during the current study are available from the corresponding author on reasonable request.

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
