# Peer review of "Pesticide Exposure of Residents Living in Wine Regions: Protocol and First Results of the Pestiprev Study"

_ijerph, 2023, doi:10.3390/ijerph20053882_

Round 1

Reviewer 1 Report

Dear authors,

Thank you for this submission. Very interesting manuscript, specially on the use of wipes to measure pesticides in different matrixes. I attach a word document with my recommendations for improvement.

Author Response

We firstly thank all the reviewers for their valuable comments and suggestions. We made changes to the article in an effort to make it more intelligible. The modifications are described hereafter, and we give a point-by-point response to the reviewers’ concerns.

Response to reviewer #1

Abstract:

  1. “It will be applied on a larger scale in 2021” -> the protocol was applied on a larger scale in 2021.

We modified the initial sentence “the protocol will be was applied on a larger scale in 2021”.

Keywords:

  1. Consider not repeating keywords that are already in the title.

We replaced the two terms “pesticides” and “residents” by the keywords: “fungicides”, “plant protection products”, “indoor pollution”.

Introduction:

  1. There is lack of references on the introduction. The authors say “determinants are poorly known” and then do not have references to studies looking at these determinants. Please consider the following recent references to support your statements: Line 38 to 43 -> “… resulting from aerial emissions of pesticides from the fields during application but also in the days after.” (Figueiredo et al. 2021, Veludo et al. 2022)
  • Figueiredo, D. M., Duyzer, J., Huss, A., Krop, E. J. M., Gerritsen-Ebben, M. G., Gooijer, Y., & Vermeulen, R. C. H. (2021). Spatio-temporal variation of outdoor and indoor pesticide air concentrations in homes near agricultural fields. Atmospheric Environment, 262, 118612. https://doi.org/https://doi.org/10.1016/j.atmosenv.2021.118612.
  • Veludo, A. F., Martins Figueiredo, D., Degrendele, C., Masinyana, L., Curchod, L., Kohoutek, J., Kukučka, P., Martiník, J., Přibylová, P., Klánová, J., Dalvie, M. A., Röösli, M., & Fuhrimann, S. (2022). Seasonal variations in air concentrations of 27 organochlorine pesticides (OCPs) and 25 current-use pesticides (CUPs) across three agricultural areas of South Africa. Chemosphere, 289, 133162. https://doi.org/https://doi.org/10.1016/j.chemosphere.2021.133162

This statement comes from the results of a systematic review published in the journal The Science of the total environment in 2021 and already quoted (reference [7] in the text). To clarify this point we moved this reference directly at the end of the quoted sentence “[…] determinants are poorly known [7]”.

Materials and methods:

  1. Line 68, the authors state “1/ The houses had to be located in the Bordeaux region, on the edge of one or several vineyards treated with pesticides;”. Given that this is a protocol I expect a more definite distance then “edge”. Was this 5 meters max from a field? 10 meters? What was the max distance allowed?

The houses and their garden adjoined plots of vineyards (i.e. no other buildings or fields between the vineyards and the dwellings). No minimal or maximal distance were required. We changed the sentence as follows: “1/The houses had to be located in the Bordeaux region, on the edge of adjoining one or several vineyards treated with pesticides (no minimum or maximum distance required)”.

  1. Line 77 -> “visit during peak spraying season”.  What do the authors define as peak spraying season? Some pesticides are only sprayed in early spring, while others are sprayed almost all here. I assume by “peak” the authors refer to most  frequent pesticide application period. Would be good to make this more clear.

We agree and make the change required: “An investigator then called the volunteers to provide detailed information about taking part and to schedule a half-day home visit during the most frequent pesticide application period peak spraying season”.

  1. Section 2.2.1 -> This questionnaire was created from scratch? How was it validated? Is it similar to other studies. For example, I know of other studies that developed similar questionnaires and were performed before PESTIPREV, such as the OBO study (referenced already by the authors), HBM4EU project, amongst others.

The main objective of the questionnaire we used was to collect information on the potential determinants of pesticide exposure (related to the local environment, the house, the occupants). This questionnaire was not designed to assess pesticide contamination or exposure which were estimated by measuring pesticides on surfaces, pets and individuals. This questionnaire was especially made for the PESTIPREV study, given its objectives and particularities (population, agricultural context). As this questionnaire was administered by a member of the research team, it was possible to reformulate the questions and to give supplemental information when needed.  

To clarify, we deleted the first sentence of the part 2.2.1. and we added the following elements to the paragraph : “The questionnaires, designed for the PESTIPREV study, recorded the characteristics of the household members […]”. 

  1. Section 2.2.2: In lines 98 to 100 the authors refer to one study on carpet dust done in the US. I would had one more reference that was performed in Europe (Netherlands) and corroborates the author statement “We hypothesized that, … these surfaces were protected from sunlight and microbial degradation, and were likely to accumulate particles over a prolonged period.” (I would had Figueiredo et al. 2022)
  • Figueiredo, D. M., Nijssen, R., J M Krop, E., Buijtenhuijs, D., Gooijer, Y., Lageschaar, L., Duyzer, J., Huss, A., Mol, H., & C H Vermeulen, R. (2022). Pesticides in doormat and floor dust from homes close to treated fields: Spatio-temporal variance and determinants of occurrence and concentrations. Environmental pollution (Barking, Essex : 1987), 301, 119024. https://doi.org/10.1016/j.envpol.2022.119024

The suggested reference was added.

  1. Section 2.2.3: The authors say the samples were stored at -18C prior to analysis. I would suggesting adding a reference or more that indicate this storage temperature is adequate.

The protocol we followed for the wipes (surface and pet wipes, patch) and the waters comes from the “Guidance document for the conduct of studies of occupational exposure to pesticides during agricultural application”, published by the OCDE in 2002. No particular temperature is documented but it is specified that: “All exposure samples should be stored temporarily in cool boxes containing dry ice or ice packs as necessary. (N.B. Dry ice can crack glass or plastic when in direct contact.) They should be transferred to a deep-freeze as soon as possible. The times of sample collection and deposition in the deep-freeze should be recorded (as should all other essential sampling and storage information) and included in the Study File”.

The following reference was added:

  • OECD Guidance Document for the Conduct of Studies of Occupational Exposure to Pesticides During Agricultural Application; Organisation for Economic Co-operation and Development: Paris, 2002.
  1. Section 2.3.1 -> Table 1 -> I am ok with this table, but I suggest adding the DT50 interval instead of the average.

We used the information on DT50 provided by the Pesticide Properties Database (PPDB) developed by the Agriculture & Environment Research Unit (AERU) at the University of Hertfordshire. This database does not provide intervals but only the average value in the case several values are available. 

  1. Section 2.3.2 -> This section as No references. I expect the lab analysis follow certain protocols or previously used methods that are presented in peer-reviewed literature. If this is a novel approach than it needs more detail and needs to be mentioned that it is a novelty. If it is not, then please reference the extraction method and also the quantification method.

This protocol was specially designed for the need of the PESTIPREV study. Details and references were added in a supplemental file (Supplementary material 1). 

  1. Section 2.4 -> Which R version? (and please reference R).
  • R Core Team (2017). R: A language and environment for statistical computing. R Foundation for Statistical Computing, Vienna, Austria. URL https://www.R-project.org/.

We specified the version (version 4.1.2) in the text and added the suggested reference. 

  1. Lines 204-207 need further explanation. The authors state “Pesticide amounts below the detection limit or not calculated due to analytical interferences were considered missing data. For pesticides that were detected but not quantified, we imputed a surface loading value equal to half the limit of quantification divided by the surface area.” -> So do the authors mean that they only imputed values that were between LOD and LOQ? And if so, why not a multiple imputation per pesticide to account for the distribution (e.g. MICE package in R)? The choice of diving LOQ by half will not reflect the distribution will reflect most likely a truncated distribution. This part needs better explanation.

Considering the small number of data below the LOD (n=3) or missing values due to interferences (n=3), we chose not to consider this data in the analysis. We specified the effectives in the text to clarify this approach.

We chose a substitution method because it appeared to be appropriate for a simple descriptive analysis of the data. Moreover, as surfaces had different dimensions, this did not strongly affect the distribution of the concentration data (we did not observed a peak of density for the values equal to LOQ/2). We added the following sentence to clarify this point: “As the matrices presented different surface areas, this substitution method was not expected to strongly affect the distribution of the data”.

Results:

  1. My main problem with results is that the data is only from three homes. A lot of different samples were collected per home, but not sure how representative are three homes. I propose the authors re-write the results as more of “Example results” and to show the power that you can get by using this protocol. I don’t think interpretation of results makes sense, it should  be more about what can we get out of a study if we use this protocol…but not on interpretation of results.  In short, I propose to send to supplement tables 3 and 4 and figure 1.

If the general objective of the PESTIPREV study is to identify the main determinants of exposure, the objectives of this pilot study were to describe the protocol implemented, the first results obtained, and the potential improvements identified for its deployment on a larger scale. For these objectives, a measurement campaign including three experimental sites appeared appropriate. Of course, three households are not sufficient to analyze the influence of factors related to agricultural practices, meteorological and topographical conditions, the layout of buildings, and residents’ characteristics and behaviors on pesticide exposure. The finality of the descriptive analysis we performed was modestly to build hypotheses on the potential determinants of residential contamination by agricultural pesticides, to be confirmed in the second phase of the study.

In order to clarify the objectives of this pilot study, we reformulate the sentence l66-69 as follows.

“The objectives of this work were: 1/ to present the protocol implemented for this pilot study, 2/ to describe the first results obtained, and 3/ to identify the potential improvements for its deployment on a larger scale in a second phase”.

As explained by the reviewer in its overall remark, for the aim of the paper, being “protocol” and “first results”, the results part is quite extensive. It appear that tables 3 and 4 are essential for us to build hypothesis on the role of potential determinants (related to cleaning habits, location, …) but Figure 1 seems perhaps redundant with the all content of the article, so we propose to moved it in supplemental files. We deleted the figure 1 and made the following changes lines 260-262:

“Figure 1 represents A figure representing the frequency of quantification per pesticide and the median surface loading for each home according to the type of surface sampled is provided in supplemental files”.

We also updated the numeration of figures consequently.

  1. Section 3.2.1 -> Table 3 -> You mention All dwellings (n=214). But there are only three dwellings, so I would not say All dwellings (n=214). I would say All surface wipes (n=214). Moreover, in this table, for some pesticide quantification frequency is below 50%, so how was it possible to calculate a median? The same for the quartiles? If this is using imputed data I would then make it clear when referencing table 3

These statistics have been calculated only for the quantified pesticides (we did not use imputed data). In order to clarify, we changed the sentence line 254 as follows: “The frequency of quantification by pesticide and the surface loadings for the quantified pesticides for the three houses are presented in Table 3 (for the quantified pesticides only)”. 

Discussion:

  1. Section 4.2 -> There is a statement here that it is false “Our data are difficult to compare with a previous observational study of residents because they did not sample the pesticides we included …” -> I see that Oerlemans et al. 2021 also measure tebuconazole using handwipes in residents:
  • Oerlemans, A., Figueiredo, D. M., Mol, J. G. J., Nijssen, R., Anzion, R. B. M., van Dael, M. F. P., Duyzer, J., Roeleveld, N., Russel, F. G. M., Vermeulen, R. C. H., & Scheepers, P. T. J. (2021). Personal exposure assessment of pesticides in residents: The association between hand wipes and urinary biomarkers. Environmental Research, 199, 111282. https://doi.org/https://doi.org/10.1016/j.envres.2021.111282

In this paragraph, we wanted to compare the results we obtained from wipe sampling on residential surfaces to the results available in the scientific literature (Oerlemans and al., have used wipes but on hands). It appeared to us that we incorrectly formulated the sentence quoted by the reviewer. So, we made the following changes:

“Our data are difficult to compare with a previous observational study of residents because they did not sample the pesticides we included and rarely used wipe sampling as a way of measuring exposure”. è “Previous observational studies rarely sampled the pesticides we included on residential surfaces using wipe sampling”. 

  1. Section 4.3 -> The studied pesticides are current-use pesticides, so the Nishioka et al. reference is not valid here. I would suggest studies on pesticides in dust from doormats or studying doormat as a determinant, such as Figueiredo et al. 2022 and Harley et al. 2019
  • Figueiredo, D. M., Nijssen, R., Krop, E. J., Buijtenhuijs, D., Gooijer, Y., Lageschaar, L., ... & Vermeulen, R. C. (2022). Pesticides in doormat and floor dust from homes close to treated fields: Spatio-temporal variance and determinants of occurrence and concentrations. Environmental Pollution, 301, 119024.
  • Harley, K. G., Parra, K. L., Camacho, J., Bradman, A., Nolan, J. E. S., Lessard, C., Anderson, K. A., Poutasse, C. M., Scott, R. P., Lazaro, G., Cardoso, E., Gallardo, D., & Gunier, R. B. (2019). Determinants of pesticide concentrations in silicone wristbands worn by Latina adolescent girls in a California farmworker community: The COSECHA youth participatory action study. Science of the Total Environment, 652, 1022–1029. https://doi.org/10.1016/j.scitotenv.2018.10.276

 The suggested articles are very interesting and investigate some of the potential factors favoring or limiting to the track-in phenomenon such as doormats. The presence of doormat is indeed a possible way to reduce the transfer of agricultural pesticides inside the home. But, we preferred to quote the article of Nishioka et al., 2001 which illustrates the track-in phenomenon more extensively. For that purpose, the authors performed repeated measures in 11 occupied and two unoccupied homes, in several rooms, both before and after lawn application of the herbicide 2,4-D, to determine transport routes of this pesticides. They were able to measure a residue concentration gradient from the entry point of the dwelling. If 2,4-D was not used on agricultural fields, but on the lawn, the results regarding the existence of a “track-in pathway” for pesticide intrusion indoors could nevertheless be similar for the agricultural pesticides.

  1. In line 452 it is stated “Therefore, no solid conclusions can be drawn from these first results. They have, however, allowed us to construct hypotheses that will have to be confirmed in the second phase of the PESTIPREV study.” -> This is what I mean with my comments about results. I think this statement is quite important and should also be in the abstract.

This point was addressed in point #13. Several modification was made consequently (reformulation of the objectives of this work, simplification of the presentation of the results). In the abstract, we chose not to present results about the potential determinants of pesticide contamination and exposure (surface location, cleaning habits, track-in, activities performed by the occupants) to avoid over-interpretation of the results. Only the results from the complete study will enable us to study the determinants of pesticide exposure. It is why we added the following sentence in the abstract: “It will be was applied on a larger scale in 2021 to study the determinants of pesticide exposure.”

Reviewer 2 Report

General comments:

People living near agricultural areas are more likely to be exposed to agricultural pesticides and suffer higher health risks than the general population. However, very little is known about the determinants of residential pesticide exposure. In order to further understand the determinants of exposure, this study first conducted an observational study in Bordeaux, France, and the preliminary results of this study are conducive to the subsequent research on the factors of residential pesticide exposure in large-scale areas. And finally propose mitigation measures to reduce pesticide exposure for people living in agricultural areas. Overall, this article is well-written and informative, but there are still some small problems that need to be improved before publication in this journal. The details are as follows:

Specific comments:

1.     Line 67: In the preliminary study, three families were selected as research objects. Is the number too small? Why did you choose three families instead of five?

2.     Line 92: Why did you choose to collect the sample by wiping it with a wet tissue?

3.     Line 96, 103: “Inside the home, we sampled surfaces considered to reflect indoor pesticide contamination over several weeks/months.” After weeks/months of sampling inside the house, were samples taken outside at the same frequency?

4.     Line 99-101: “… were likely to accumulate particles over a prolonged period.” “shorter-term exposure”——Samples were collected from surfaces that may be exposed to pesticides in the long term and the short term, but the difference analysis of pesticide exposure in the following two periods seems to be insufficient, and it is hoped that some supplements can be made.

5.     Line 193: Please confirm “…LOQ pyraclostrobin=0.4 ng/L]” If the parentheses here are redundant.

6.     Line 224-226: Houses 1 and 3 had discontinuous hedgerows separating the garden from the vineyard, while House 2 had no hedgerows, which influenced whether the three houses were exposed to pesticides.

7.     Line 251: The explanation of Figure 1 seems too simple. Could you make some additions? For example, the concentration difference of pesticides between three households.

8.     Line 264: Please check “... in Table 4 and represented in.” Is there a format issue here.

9.     Line 274: Higher levels of benalaxyl, tebuconazole and trifloxystrobin were observed in the patches and hand washing water of children who played with dogs. Could it be further explored whether this is a factor of dogs?

10.   Line 18, 21, 182, 259, 278-280: Please check whether the format between “ng” and “ng/L” and numbers is correct. ((LOQ = 0.7ng), (LOQ = 1.0 ng), ((LOQ = 0.7ng/L) …)

11.   Line 381: "Our study is complementary to other studies on this subject." Could you give a few examples of other people's research to better illustrate what it adds?

12.   Line 415-416: “The high dose of this pesticide applied per hectare or its long half-life could be explanatory factors in this case.” When it is explained that the surface load of bactericidal agent is related to high dose application or half-life, some relevant data or literature support can be appropriately added.

13.   Line 441: What does “2, 4-D” mean?

14.   This study is mainly aimed at exploring the determinants of residents' pesticide exposure, but it seems that there is no specific description of these exposure factors. For example, how do these factors produce exposure to the population or what are the main ways that residents are exposed to pesticides.

15.   This paper spent a lot of space to explain and improve the questionnaire survey, but there is little analysis of residents' exposure factors. Could you please make some additions? For example, what are the differences in the levels of pesticide detected in the three families, what are the main influencing factors of pesticide exposure, and whether there are significant differences between the levels of pesticide detected in each family, etc.

Author Response

We firstly thank the reviewers for their valuable comments and suggestions. We made changes to the article in an effort to make it more intelligible. The modifications are described hereafter, and we give a point-by-point response to the reviewers’ concerns.

Response to reviewer #2 

  1. Line 67: In the preliminary study, three families were selected as research objects. Is the number too small? Why did you choose three families instead of five?

This remark is very similar to the point #13 made by the first reviewer. Three households are indeed not sufficient to analyze the determinants of pesticides exposure, but was enough to test our protocol. The finality of the descriptive analysis we performed was modestly to build hypotheses on the potential determinants of residential contamination by agricultural pesticides, to be confirmed in the second phase of the study (2021).

  1. Line 92: Why did you choose to collect the sample by wiping it with a wet tissue?

According to the US-EPA, wipe sampling is an important technique for the estimation of contaminant deposition in buildings, homes, or outdoor surfaces as a source of possible human exposure. This method has already been performed to measure pesticides in previous works. The wipes are wetted with a solvent to remove contaminants from the sampled surfaces. The nature of the solvent must be adapted according to the pollutant targeted and the media.

  • U.S. EPA A Literature Review of Wipe Sampling Methods for Chemical Warfare Agents and Toxic Industrial Chemicals. 2007, EPA/600/R-11/079, 59.
  1. Line 96, 103: “Inside the home, we sampled surfaces considered to reflect indoor pesticide contamination over several weeks/months.” After weeks/months of sampling inside the house, were samples taken outside at the same frequency?
  2. Line 99-101: “… were likely to accumulate particles over a prolonged period.” “shorter-term exposure”——Samples were collected from surfaces that may be exposed to pesticides in the long term and the short term, but the difference analysis of pesticide exposure in the following two periods seems to be insufficient, and it is hoped that some supplements can be made.

As stipulated in part 2.1, we only did one visit per dwelling. Therefore, all the surface samples were collected at the same moment. We chose to collect pesticides from surfaces representing different temporalities of the contamination and different potential way of exposure. We indeed made several hypotheses in part 2.2.2:

  • Dusted high horizontal surfaces would reflect indoor pesticide contamination over several weeks/months because they are likely to accumulate particles over a prolonged period. People could be exposed all the year to pesticides in house dust due to resuspension of particles and dust.
  • Other surfaces (floors, furniture) would might reflect shorter-term exposure.
  • Frequently touched surfaces (light switches, decorative objects, toys and digital equipment) might reflect the handling of pesticides.
  • Outdoor surfaces would reflect the most recent local deposits of pesticides and the potential exposure to pesticides during outdoor activities.
  1. Line 193: Please confirm “…LOQ pyraclostrobin=0.4 ng/L]” If the parentheses here are redundant.

The extra-parenthesis was deleted.

  1. Line 224-226: Houses 1 and 3 had discontinuous hedgerows separating the garden from the vineyard, while House 2 had no hedgerows, which influenced whether the three houses were exposed to pesticides.

Yes, it is correct. We reformulated the sentence as follows: “Houses 1 and 3 were located on flat land and had discontinuous hedgerows separating the garden from the nearest vineyards. Dwelling 2 was located on a hillside and had no hedgerow”.

  1. Line 251: The explanation of Figure 1 seems too simple. Could you make some additions? For example, the concentration difference of pesticides between three households.
  2. Line 264: Please check “... in Table 4 and represented in.” Is there a format issue here.

As requested by the first reviewer, this figure was moved in supplemental files. Modifications were made consequently.

  1. Line 274: Higher levels of benalaxyl, tebuconazole and trifloxystrobin were observed in the patches and hand washing water of children who played with dogs. Could it be further explored whether this is a factor of dogs?

As developed in the response of the first remark, three households are not sufficient to analyze the determinants of pesticide exposure. The finality of the descriptive analysis we performed was modestly to build hypotheses on the potential determinants of residential contamination by agricultural pesticides, to be confirmed in the second phase of the study. Several modification were made according to the reviewers to clarify this point (reformulation of the objectives, simplification of the presentation of the results, clarification of the perspectives in the abstract).

  1. Line 18, 21, 182, 259, 278-280: Please check whether the format between “ng” and “ng/L” and numbers is correct. ((LOQ = 0.7ng), (LOQ = 1.0 ng), ((LOQ = 0.7ng/L) …)

The numbers provided in the article have been checked and are correct.

  1. Line 381: "Our study is complementary to other studies on this subject." Could you give a few examples of other people's research to better illustrate what it adds?

To add clarity, we moved the references of other studies focusing on the pesticide exposure in population living near fields at the end of the sentence: “These studies typically include fewer environmental measurements per household and often focus on indoor air or house dust collected with a vacuum cleaner [23–25]”. Three important European studies are quoted:

The OBO study (Netherlands)

  • Figueiredo, D.M.; Krop, E.J.M.; Duyzer, J.; Gerritsen-Ebben, R.M.; Gooijer, Y.M.; Holterman, H.J.; Huss, A.; Jacobs, C.M.J.; Kivits, C.M.; Kruijne, R.; et al. Pesticide Exposure of Residents Living Close to Agricultural Fields in the Netherlands: Protocol for an Observational Study. JMIR Research Protocols 2021, 10, doi:10.2196/27883.

Several papers from this study are referenced in our manuscript.

The sprint STUDY (10 EU case study sites and one site in Argentina, https://sprint-h2020.eu/)

  • Silva, V.; Alaoui, A.; Schlünssen, V.; Vested, A.; Graumans, M.; van Dael, M.; Trevisan, M.; Suciu, N.; Mol, H.; Beekmann, K.; et al. Collection of Human and Environmental Data on Pesticide Use in Europe and Argentina: Field Study Protocol for the SPRINT Project. PLoS One 2021, 16, e0259748, doi:10.1371/journal.pone.0259748.

The PESTIRIV study (France)

  • Santé publique France PestiRiv : Étude d’exposition aux pesticides chez les riverains de zones viticoles et non viticoles. Protocole. 2021, 83.
  1. Line 415-416: “The high dose of this pesticide applied per hectare or its long half-life could be explanatory factors in this case.” When it is explained that the surface load of bactericidal agent is related to high dose application or half-life, some relevant data or literature support can be appropriately added.

This is a hypothesis we made to explain the relative high surface loadings we observed. The indicative values of dose applications and half-lives of the pesticides were provided in Table 1 and the references in the section “Methods”.

  1. Line 441: What does “2, 4-D” mean?

We specified in the text: 2,4-dichlorophenoxyacetic acid (2,4-D)

  1. This study is mainly aimed at exploring the determinants of residents' pesticide exposure, but it seems that there is no specific description of these exposure factors. For example, how do these factors produce exposure to the population or what are the main ways that residents are exposed to pesticides.
  2. This paper spent a lot of space to explain and improve the questionnaire survey, but there is little analysis of residents' exposure factors. Could you please make some additions? For example, what are the differences in the levels of pesticide detected in the three families, what are the main influencing factors of pesticide exposure, and whether there are significant differences between the levels of pesticide detected in each family, etc.

The main objective of the descriptive analysis we performed was modestly to build hypotheses on the potential determinants of residential contamination by agricultural pesticides, to be confirmed in the second phase of the study. According to the results presented in part 3.2 and 3.3, in addition to analytical performance, certain parameters seemed to influence the pesticide concentrations in the home, such as the quantities of pesticides sold in the area, the type of surface sampled, the room itself, and the cleaning habits. These results were discussed in part 4.3. The role of these factors will be extensively explored in the second part of the study using adequate statistical models (a new measurement campaign was realized in 2021).

Reviewer 3 Report

The article by Teysseire et al., entitled “Pesticide Exposure of Residents Living in Wine Regions: Protocol and First Results of The Pestiprev Study” is characterizing human exposure to pesticides using several non-conventional sampling techniques (e.g. wipes, skin’s patches, hand-washing, pets). The sampling design is appropriate. Overall, the article is interesting and provide valuable information for the scientific community. Although it is a pity that only 6 pesticides were targeted, the high amount of samples analyzed make this article worth to be published in this journal. Additional details should be provided on the analytical methods used. In addition, there are several issues detailed below worth to be considered by the authors.

l.30-37: Whenever possible, consider to be more precise than just pesticides and specify which classes for which the arguments are valid (e.g. organophosphates, pyrethorids, neonicotinoids).

l.68: Precise how far is the edge

l.263: This section misses many relevant information that are usually in an article, e.g. the internal standards used, the type of LC-MS/MS used, type of column used, the instrument programme …

l.247-248: How far are these two houses from each other? Could it be different pesticides applied in the close fields?

All Figures: Consider to increase the font size for better readability (particularly for the x axis labels).

Figure 1: The median line is not visible for the first box (outdoor). If possible, update it.

Figure 2: If you would display this Figure separately for the different houses, would the results (in terms of variations in the concentrations) be the same?

l.351-353: I would suggest to nuance this statement, as with a low amount of households, nothing can be concluded on the social determinants of exposure to pesticides.

l.373-376: You could also consider using some apps on smartphones collecting such information with less burden for the participants.

l.400-405: It is a pity that the instrumental limits of quantification are not available in the mentioned study as the fact that the mentioned pesticides are quantified more frequently in wipes than in dust could suggest that the importance of dermal contact in comparison with dust ingestion might have been underestimated.

l.411-416: I would suggest to the authors to nuance this statement, taking into account the importance of the environmental fate of pesticides (more than quantity of pesticides applied) driven by their physico-chemical properties on the concentrations of pesticides in various environmental media.

l.420-421: It could also suggest some indoor sources from materials or products containing pesticides.

Section 4.1: The authors can consider to place it as the last section of discussion rather than at the beginning. The authors could also provide a recommendation on the ideal surface to sample with wipes, its height, and the relevance for characterizing human exposure.

Minor details:

l. 18 (and in many other places): add a space before ng

l.26: replace will be by was

l.49: I do not think levers is adequate here. Consider to modify.   

l.115: correct field

l.264. remove and are represented in

l.334 (and elsewhere): values is not appropriate, replace by levels or concentrations

l.476: Some interesting typo to correct

Author Response

We firstly thank the reviewers for their valuable comments and suggestions. We made changes to the article in an effort to make it more intelligible. The modifications are described hereafter, and we give a point-by-point response to the reviewers’ concerns.

Response to reviewer #3

  1. 30-37: Whenever possible, consider to be more precise than just pesticides and specify which classes for which the arguments are valid (e.g. organophosphates, pyrethorids, neonicotinoids).

Regarding the observational studies, we were able to specify the classes of pesticides for which the arguments are valid (organophosphates, organochlorines, pyrethroids).

Regarding the epidemiological studies, most of the publications provided conclusions all pesticides considered and it was not possible to precise the classes of substances.

  1. 68: Precise how far is the edge

The houses and their gardens adjoined plots of vineyards (i.e. there were no other buildings or fields between the vineyards and the dwellings). No minimal or maximal distance were required. We changed the sentence as follows: “1/The houses had to be located in the Bordeaux region, on the edge of adjoining one or several vineyards treated with pesticides (no minimum or maximum distance required)”.

The distance between the home and the nearest field was reported in Table 2 (respectively 18, 7 and 5 meters for dwellings 1, 2 and 3).

  1. 263: This section misses many relevant information that are usually in an article, e.g. the internal standards used, the type of LC-MS/MS used, type of column used, the instrument programme …

Extensive details on the protocol and relevant references were added in supplemental files (supplementary material 1).

  1. 247-248: How far are these two houses from each other? Could it be different pesticides applied in the close fields?

The first dwelling was more than 30 km away from the others. The second and third dwellings were about 10 km away. It is therefore highly probable that the phytosanitary treatments applied on the fields bordering the dwellings differed in their nature, intensity and date of application.

  1. All Figures: Consider to increase the font size for better readability (particularly for the x axis labels).

Figures were modified.

  1. Figure 1: The median line is not visible for the first box (outdoor). If possible, update it.

The figure was updated.

  1. Figure 2: If you would display this Figure separately for the different houses, would the results (in terms of variations in the concentrations) be the same ? 

In order to observe the influence of the room on the pesticide concentrations, we decided to present the results for all the houses combined. The number of samples collected per room and per house being rather low, it is sometimes difficult to analyze these same results per dwelling. Nevertheless, some trends are confirmed on the graphical representations by household. For example, we still observe higher concentrations in the main entrance of the house for most of the pesticides, and concentrations lower outside than in many rooms.

  1. 351-353: I would suggest to nuance this statement, as with a low amount of households, nothing can be concluded on the social determinants of exposure to pesticides.

We reformulated the sentence as follows: “First, the method of selecting participants should be adapted. Identifying eligible housing in the area using a Geographic Information System (GIS) and then making contact by phone or letter could enable us to include a wide range of profiles”.

  1. 373-376: You could also consider using some apps on smartphones collecting such information with less burden for the participants.

This is an interesting suggestion, which we did not think of for the new study we have already conducted in 2021.

  1. 400-405: It is a pity that the instrumental limits of quantification are not available in the mentioned study as the fact that the mentioned pesticides are quantified more frequently in wipes than in dust could suggest that the importance of dermal contact in comparison with dust ingestion might have been underestimated.

This is an interesting remark. Authors did not discuss this point in their publication.

  1. 411-416: I would suggest to the authors to nuance this statement, taking into account the importance of the environmental fate of pesticides (more than quantity of pesticides applied) driven by their physico-chemical properties on the concentrations of pesticides in various environmental media.

In this first descriptive analysis, we observed that “the surface loadings seem to be better explained by the quantities of pesticides sold in Gironde in 2019”. This statement did not exclude some other potential factors and we totally agree with the remark that the physico-chemical properties strongly influence the concentrations of pesticides found in various environmental media. We are currently studying the effect of these physico-chemical properties on measured pesticide concentrations, based on the new data we collected in 2021, using mechanistic modeling of pesticide emission and dispersion.

  1. 420-421: It could also suggest some indoor sources from materials or products containing pesticides.

In the part 2.3.1, we specified that the pesticides we searched for “are not intended for domestic use, except for tebuconazole, which is also used to protect wood against wood-eating insects”.

  1. Section 4.1: The authors can consider to place it as the last section of discussion rather than at the beginning. The authors could also provide a recommendation on the ideal surface to sample with wipes, its height, and the relevance for characterizing human exposure.

As testing the validity of our protocol was the first objective of this pilot study, we indeed chose to conclude on this point at the beginning of the part 4.1.

According to the US-EPA, wipe sampling is an important technique for the estimation of contaminant deposition in buildings, homes, or outdoor surfaces as a source of possible human exposure. This method has already been performed to measure pesticides in previous works. The wipes are wetted with a solvent to remove contaminants from the sampled surfaces. The nature of the solvent must be adapted according to the pollutant targeted and the media. In our case, wipe sampling is an interesting tool to characterize the pesticide contamination of indoor and outdoor environments, and can help us determining the external exposure of the residents living near fields. To do so, we chose to collect pesticides from surfaces representing different temporalities and modalities of the contamination and the exposure. We made several hypotheses developed in the first paragraph of section 2.2.2:

  • Dusted high horizontal surfaces would reflect indoor pesticide contamination over several weeks/months because they are likely to accumulate particles over a prolonged period. People could be exposed all the year to pesticides in house dust due to resumption of particles and dust.
  • Other surfaces (floors, furniture) would might reflect shorter-term exposure.
  • Frequently touched surfaces (light switches, decorative objects, toys and digital equipment) might reflect the handling of pesticides.
  • Outdoor surfaces would reflect the most recent local deposits of pesticides and the potential exposure to pesticides during outdoor activities.

It is difficult to determine numerically the minimum dimensions of the surfaces to sample. Our finding (“Improvements could be made by avoiding sampling porous or rough surfaces and focusing on flat, geometrically well-defined surfaces. Sampling of very small areas should also be avoided, as this greatly increases the sensitivity of the measurement”) it is the result of our field observations and remains mostly empirical.

Minor details:

  1. 18 (and in many other places): add a space before ng

A space was added l18.

  1. 26: replace will be by was

We modified the initial sentence “the protocol will be was applied on a larger scale in 2021”.

  1. 49: I do not think levers is adequate here. Consider to modify.

We replaced action levers by measures as follows: “This would also contribute to identifying measures to mitigate the exposure of populations living in agricultural areas”.

  1. 115: correct field

Filed blanks” was corrected as “Field blanks”.

  1. 264. remove and are represented in

As requested by the first reviewer, we moved the figure 1 in supplemental files and made the following changes: “Figure 1 represents A figure representing the frequency of quantification per pesticide and the median surface loading for each home according to the type of surface sampled is provided in supplemental files”. 

  1. 334 (and elsewhere): values is not appropriate, replace by levels or concentrations

The term “value” was replaced by “surface loading” and “level” in the text.

  1. 476: Some interesting typo to correct

We deleted the section “6. Findings” because a Funding section was already present.